chemical physics/materials science/ crystallography

polymer, irradiation, crystallization, spherulite, beta-phase

**Author for correspondence:**
Petr Svoboda
e-mail: svoboda@utb.cz

This article has been edited by the Royal Society of Chemistry, including the commissioning, peer review process and editorial aspects up to the point of acceptance.

# Study of crystallization behaviour of electron beam irradiated polypropylene and high-density polyethylene

Petr Svoboda[1], Krunal Trivedi[1], Karel Stoklasa[1], Dagmar Svobodova[2] and Toshiaki Ougizawa[3]

[1]Department of Polymer Engineering, Faculty of Technology, Tomas Bata University in Zlin, Vavreckova 275, 762 72 Zlin, Czech Republic
[2]Faculty of Humanities, Tomas Bata University in Zlin, Stefanikova 5670, 760 01 Zlin, Czech Republic
[3]Department of Organic and Polymeric Materials, Tokyo Institute of Technology, 2-12-1-S8-33, Ookayama, Meguro-ku, Tokyo 152-8552, Japan

PS, 0000-0002-7320-5467; KT, 0000-0001-9389-9737; KS, 0000-0003-3947-6447; DS, 0000-0001-5512-4965; TO, 0000-0002-7761-6909

The influence of electron-beam irradiation on polypropylene (PP) and high-density polyethylene (HDPE) was investigated with a focus on crystallization. A high-temperature (200°C) creep test revealed that the HDPE gradually increased cross-linking density in the range 30–120 kGy, while the PP underwent a chain scission which was quantitatively evaluated by gel permeation chromatography. The mechanical properties were measured in the range -150 to 200°C by a dynamic mechanical analysis. A small presence of C=C and C=O bonds was found in the irradiated PP by a Fourier transform infrared spectroscopy. Crystallization kinetics measured by differential scanning calorimetry and hot-stage optical microscopy results were influenced tremendously by irradiation for HDPE and to a lesser extent for PP. Irradiation caused a decrease in both the number of nucleation centres and the growth rate of individual spherulites. Crystallization was analysed in detail with the help of Hoffman–Lauritzen, Avrami and Arrhenius equations. Interestingly an increasing β-crystal formation with an increasing irradiation level was discovered for PP by X-ray diffraction. A generation of defects in the crystalline structure owing to irradiation was discussed.

## 1. Introduction

Polypropylene (PP) and polyethylene (PE) are the most widely used polymer materials in pure form but also modified in various ways

[1–3]. The molecular structure of polymers and consequently their properties can be significantly modified by electron beam irradiation. The main reactions during the irradiation process are chain scission, chain branching and cross-linking. Usually, all these reactions coexist and it is important to have detailed knowledge of the influence of the radiation level on the property change [4]. The effect, which predominates, depends on several factors, such as the chemical structure and morphology of the polymer as well as the irradiation conditions and the post-treatment. In order to predict the behaviour of carbon-chain polymers exposed to ionizing radiation, an empirical rule can be used. According to this rule, polymers containing a hydrogen atom at each carbon atom predominantly undergo cross-linking, whereas in the polymers containing a quaternary carbon atom, that is the unit $-CH_2-CR_1R_2CH_2-$, where $R_1$ and $R_2$ are groups other than H, the chain scission predominates [5]. The irradiation of neat PP without any additives despite the formation of a few branches predominantly leads to a significant decrease in the molecular weight owing to $\beta$-chain scission [6] and also to the formation of double bonds. Furthermore, the addition of free radicals to double bonds takes place and long-chain branches are formed, followed by an increase in the molecular weight. Additionally, a disproportionation or recombination reactions of two polymer radicals occur which also leads to increased molecular weight.

Irradiated PE has distinctly different coexisting chain configurations because of the polycrystalline and partially crystalline character of the system. Keller & Ungar [7] have studied the effect of radiation on crystals of PE and paraffin. They reported the destruction of the crystalline structure of PE above a certain dose where the radiation temperature approaches the temperature of orthorhombic-hexagonal transition. Crystallization can be influenced also by the addition of fillers [8]. When linear PP is modified by electron-beam (e-beam) and when the irradiation dose increases, at first very few but long branches are created, then the trend turns towards many shorter branches per molecule. This high degree of branching with small arm molar masses is typically found in low-density polyethylene (LDPE). The topography of the long branches and the comparison of linear and irradiated PP show a small degree of branching with high molar masses of the branches [9]. PP homopolymer and PP-ethylene copolymer irradiated under nitrogen atmosphere by e-beam exhibit efficient branching as a small amount of ethylene in the propylene copolymer promotes branching over degradation [10].

E-beam irradiated linear isotactic polypropylene (iPP) homopolymer irradiated at different temperatures shows not only a slight decrease in molar mass but also an increased degree of long-chain branching and increased crystallization temperature. At temperatures higher than 100°C the PP is partially molten, an amorphous phase increases and the formed branched molecules have a higher number of shorter long-chain branched (LCB) molecules. Irradiation at 210°C leads to a significant molar mass reduction. A change of molecular architecture from a slightly branched star-like type to a higher branched tree-like type was found in samples irradiated at 20 kGy [11]. The change from star-like to a tree-like branching topography is well documented. E-beam and gamma-irradiated PP undergoes a chain scission and generated macro-radicals can form LCB under appropriate irradiation conditions. It was observed that with smaller doses, the zero shear-rate viscosities $\eta_o$ of the electron beam irradiated PP were above the values for the unmodified PP with the same $M_w$, although they were distinctly lower than the linear reference with high doses [12]. Krause *et al.* [11] modified PP by e-beam irradiation, generated LCB and found that crystallization temperatures increased for annealed samples, while they decreased for non-annealed samples. Moreover, stable products were obtained only by irradiation in a nitrogen atmosphere followed by annealing. Irradiated PP in molten and in solid-state created LCB, samples irradiated at 200°C had a lower molecular weight (generated higher branched samples) than samples irradiated at 25°C. Also melting, crystallization and glass transition temperatures were reduced [13].

Crystalline structure was studied and a decreased crystallization temperature with a higher e-beam irradiation dose was reported for cross-linked LDPE and high-density polyethylene (HDPE) [14]. Interestingly, two different melting temperatures and increased fusion enthalpy were reported for e-beam irradiated HDPE. No changes in the crystallization rate or in the crystallite size were observed by X-ray scattering before and after thermal treatment of the samples [15]. Mishra *et al.* studied e-beam irradiated PP and found increased crystallinity, thermal stability and melting temperature but an unaffected isotactic structure [16]. Pawde *et al.* [17] investigated PP films after e-beam irradiation in the air. E-beam irradiation led to cross-linking which led to increased Young's modulus and to changes in the dielectric constant, decreased tensile strength and elongation at break. The irradiated PP had a higher impact strength and it could be used as an alternative to nylon rope. Electron beam irradiated PP with a dose rate of 20 kGy could be used as a good capacitor dielectric because of its very low dielectric loss and excellent dielectric strength [17]. Dielectric properties of PP films after e-beam irradiation were also studied and a slight increase in absorbance was found [18]. According to Lu *et al.* [19], the radiochemical reaction yield (G value) of the i-PP irradiated by e-beam in a vacuum

is higher for the chain scission than for cross-linking, i.e. the chain scission predominates. When irradiated in air or oxygen, iPP mainly undergoes a chain scission owing to a chain oxidation reaction. The melt index increases with the increasing dose. Low-dose irradiation of iPP (0.75 kGy) could improve the PP properties, Young's modulus increases by 172 MPa, whereas 60 kGy increased Young modulus by 210 MPa [19].

As can be seen from the references above [17,19], some contradictory findings were reported. In order to investigate the origins of the encountered discrepancies, we investigated the influence of e-beam irradiation on PP even further. To the best of our knowledge, the PP or HDPE abilities to crystallize after their exposure to the same irradiation conditions have not been investigated to date. This research aims not only to explore the crystallization behaviour of e-beam irradiated PP, but also to compare it with that of the e-beam irradiated HDPE.

## 2. Experimental

The PP with trade name C766–03 was supplied by Dow Chemical (Europe) and HDPE with trade name HTA-002 was supplied by ExxonMobil Chemical. The main characteristics of both materials are listed in table 1.

PP and HDPE sheets were prepared by compression moulding at 200°C for 6 min and at 150°C for 5 min, respectively. Beta (electron beam) irradiation was performed on PP and HDPE sheets (sample size was $12 \times 6 \times 0.6$ mm) in the air at room temperature, in BGS Beta-Gamma-Service GmbH, Germany. It was made sure that the temperature did not exceed 50°C. The source of radiation was a toroid electron accelerator Rhodotron (10 MeV, 200 kW). The irradiation was performed in a tunnel on a continuously moving conveyer with the irradiation dosage ranging from 30 to 120 kGy, in steps of 30 kGy per pass. Samples were placed in one layer and sealed between polyethylene terephthalate sheets. Other important parameters were 10 MeV, 10 mA, conveyer belt speed 3 m min$^{-1}$, distance from the scanner to sample 78 cm and irradiation time 2 s.

Tensile specimens were cut out of the non-irradiated and irradiated sheets and used for the tensile creep experiments according to ISO 899 standard. Creep behaviour was studied in a Memmert oven with a temperature control of ±2°C. Creep was recorded by a camera (Sony-SLT-A33 which had a capability of recording HD video (1920 × 1080 pixels) at 25 fps) for further analysis. The effects of a high temperature (200°C) at stress level 0.1 MPa on the creep behaviour of irradiated and non-irradiated PP and HDPE were studied.

A Perkin-Elmer DSC-1 was used to evaluate the crystallization kinetics. The Indium standard was used for temperature calibration. Nitrogen at a flow rate of 20 ml min$^{-1}$ was employed during the experiment. For the analysis of isothermal crystallization, samples were heated to 200°C (at 100°C min$^{-1}$ heating rate) and then cooled (at 50°C min$^{-1}$) to the isothermal crystallization temperature (118–135°C). In all cases, samples were held at 200°C for 5 min to eliminate any previous thermal history. From differential scanning calorimetry (DSC), in order to evaluate the relative degree of crystallinity ($X$) of irradiated and non-irradiated samples, the following equation was used:

$$X = \frac{\Delta H}{\Delta H_{100}} \times 100,$$

where $\Delta H$ is the heat of crystallization of the PP or HDPE and $\Delta H_{100}$ is the value of heat of crystallization for 100% crystalline PP or HDPE ($\Delta H_{100} = 209$ J g$^{-1}$ for PP [20] and $\Delta H_{100} = 293$ J g$^{-1}$ for HDPE [21]).

The crystallization was analysed initially by DSC and then also by polarized optical microscopy. The first step is the heating of the samples (10–11 mg) from room temperature to 200°C at the rate of 100°C min$^{-1}$ which is followed by isothermal annealing lasting 5 min to assure complete melting of PP and HDPE crystals ($T_m$ of PP is about 165°C and $T_m$ of HDPE is about 134°C). The second step was a quenching (at a cooling rate 50°C min$^{-1}$) of the sample to the desired isothermal crystallization temperature (in the range 118–136°C). This was possible by the employment of a cooling machine (capable of cooling to −130°C). The third step was the isothermal crystallization at the desired temperature (118–136°C). The time when the heat flow curve reaches the minimum value and then starts to grow to form an exothermal peak was assigned to be $t = 0$. The relative crystallinity curve was obtained by the integration of the heat flow curve. When the relative crystallinity has a value of 0.5 (or 50%), half time of crystallization $\tau_{1/2}$ is calculated. Then, the crystallization kinetics can be expressed as $\tau_{1/2}^{-1}$.

**Table 1.** Properties of pure materials.

| | polypropylene C766-03 | high-density polyethylene HTA 002 |
|---|---|---|
| melt flow rate | 3.5 g/10 min (ISO 1133) | 0.15 g 10 min$^{-1}$ (ASTM D1238) |
| flexural modulus | 1.156 GPa (ISO 178) | — |
| charpy impact strength | 10 kJ m$^{-2}$ at 23°C (ISO 179) | — |
| density | — | 0.952 g cm$^{-3}$ (ExxonMobil Method) |

The irradiated and pristine PP's spherulite growth was observed by hot-stage optical microscopy. The specimen was melt-pressed for 1–2 min between two cover glasses on a hot stage at an elevated temperature of 200°C. The melted specimen was then placed onto a LINKAM hot stage of the microscope set to a desired temperature in the range 130–140°C. An optical microscope (LMU-406 SP) equipped with a video recording system was used for structural development during the isothermal annealing.

Dynamic mechanical measurements were carried out on a dynamic mechanical analyser ITKeisoku-seigyo (DVA-200S). The samples were measured in a cyclic tensile strain mode with a frequency of 10 Hz. The heating rate was 5°C min$^{-1}$ in the temperature range −150 to 200°C.

The Fourier transform infrared (FTIR) study was carried out by the Nicolet 320 Avatar FT-IR spectrometer in ATR mode. The sheets were scanned from 4000 to 400 cm$^{-1}$ with scanning number 64.

An X-ray diffractometer, X'Pert PRO from PANalytical, was used to analyse the PP and HDPE sheets with the scanning range of 5–30° (2$\theta$). Other parameters were $U = 40$ kV, $I = 30$ mA and $\lambda = 0.154$ nm (CuK$\alpha$).

For the small angle X-ray scattering (SAXS) analysis was used by Anton Paar SAXSpace. Samples were placed in the holder, the distance between the sample and the detector was 268.5 mm. CuK$\alpha$ was used with $U = 40$ kV, $I = 50$ mA, exposition time $t = 15$ min. An imaging plate was used as a detector.

The measurements of molecular weight were done at 160°C on a Polymer Laboratories PL 220 high-temperature chromatograph (Polymer Laboratories, Varian Inc., Church Stretton, Shropshire, England) equipped with three 300 mm x 7.5 mm PLgel Olexis columns and a differential refractive index detector. 1,2,4-trichlorobenzene (TCB) was used as an eluent, stabilized with an antioxidant butylhydroxytoluene (Ciba, Basel, Switzerland). The flow rate of the mobile phase was 1 ml min$^{-1}$ and in all cases, 200 µl was injected. All samples were prepared to a concentration of 0.5 mg ml$^{-1}$ in TCB. For calibration purposes, narrowly distributed PE standards (Polymer Standards Service GmbH, Mainz, Germany) were used.

# 3. Theoretical background

## 3.1. Avrami analysis

Semicrystalline polymers melt during heating above melting temperature ($T_m$) and crystallize during cooling below $T_m$. Crystallization kinetics can be analysed using a classical Avrami equation as given in equation (3.1) [22]:

$$1 - X_t = \exp(-kt^n), \tag{3.1}$$

where $k$ is the Avrami rate constant and $n$ is the Avrami exponent. Both $k$ and $n$ depend on the nucleation and growth mechanisms of spherulites.

From the DSC isothermal crystallization measurement, the crystallinity $X_t$ can be calculated from the area of the exothermic peak at a crystallization time $t$ divided by the total area under the exothermic peak:

$$X_t = \frac{\int_0^t (dH/dt)dt}{\int_0^\infty (dH/dt)dt}, \tag{3.2}$$

where the numerator represents the heat generated at time $t$ and the denominator means the total heat generated up to complete crystallization.

In order to evaluate the Avrami constants by linear regression, equation (3.1) can be rewritten into the double logarithmic form as follows:

$$\ln[-\ln(1 - X_t)] = \ln k + n \ln t. \tag{3.3}$$

The $k$ and $n$ values are then obtained using equation (3.3) from the slope and intercept of the linear regression line.

## 3.2. Hoffman–Lauritzen analysis

Hoffman and Lauritzen studied the crystallization behaviour of the polymers and proposed an equation for the chain folded crystal growth rate $G$ [23,24]:

$$G = G_0 \exp\left[\frac{-U^*}{R(T_c - T_\infty)} - \frac{K_g}{T_c(\Delta T)f}\right], \tag{3.4}$$

where $U^*$ is a constant (1500 cal mol$^{-1}$) characteristic for the activation energy for repetitive chain motion, $R$ is the gas constant, $T_c$ is the crystallization temperature (K), $T_\infty = T_g - 30$ K (for PP the glass transition temperature $T_g = 270$ K), $\Delta T = T_m^0 - T_c$, $T_m^0$ is the equilibrium melting temperature of an infinitely thick crystal, $f$ is a correction factor and it equals to $2T_c/(T_m^0 + T_c)$, $K_g$ is the nucleation constant and $G_0$ is a pre-exponential factor. This equation is often used in the logarithmic form:

$$\ln(G) + \frac{U^*}{R(T_c - T_\infty)} = \ln G_0 - \frac{K_g}{T_c \Delta T f}. \tag{3.5}$$

## 3.3. Arrhenius equation

Svante Arrhenius recognized that the typical temperature dependence indicates an exponential increase of the rate, or rate constant, with temperature, which can be written as

$$k = A e^{-E_a/RT}, \tag{3.6}$$

where $A$ is called the pre-exponential factor, $R$ is the universal gas constant (8.314 J K$^{-1}$ mol$^{-1}$), $T$ is the absolute temperature in K and $E_a$ is the activation energy. It is often used in the logarithmic form:

$$\ln k = -\frac{E_a}{RT} + \ln A. \tag{3.7}$$

The empirical constants $E_a$ and $A$ can be deduced from the slope and intercept of a $\ln k$ versus $1/T$ plot [25]. In the temperature range 120–140°C, we observed an exponential decay of the crystallization kinetics of PP and HDPE (as opposed to the exponential increase described by equation (3.6)), the regression according to the Arrhenius equation gave us the best fit.

# 4. Results and discussion

Figure 1a,b shows high-temperature (200°C) creep results for PP and HDPE under constant stress of 0.1 MPa for various irradiation doses. In figure 1a, it is illustrated that PP samples stretched all the way at 200°C within 55 s regardless of the irradiation dose (only samples with 0 and 120 kGy are shown). Apparently, irradiation did not cause cross-linking in the case of PP. By contrast with PP, the HDPE (figure 1b) exhibits very different high-temperature creep results. Even slightly irradiated HDPE (30 kGy) demonstrates some resistance to creep. This resistance to creep gradually improves with an increasing irradiation dose. Apparently irradiation is a very effective way to cross-link PE and in the observed range of irradiation 30–120 kGy the improvement of cross-linking is gradual. As a result, we have two irradiated samples, PP and HDPE, that could show quite different crystallization behaviour. Consequently, this detailed crystallization behaviour study for two very different materials is the main subject of this paper.

For the cross-linked PE the measurement of molecular weight was not possible. However, for the irradiated PP, that was freely flowing at an elevated temperature, we were able to evaluate the irradiation effect on molecular weight distribution by gel permeation chromatography (GPC) (figure 2 and table 2). As reported by the majority of researchers, our PP sample underwent chain scission that resulted in a decrease in molecular weight.

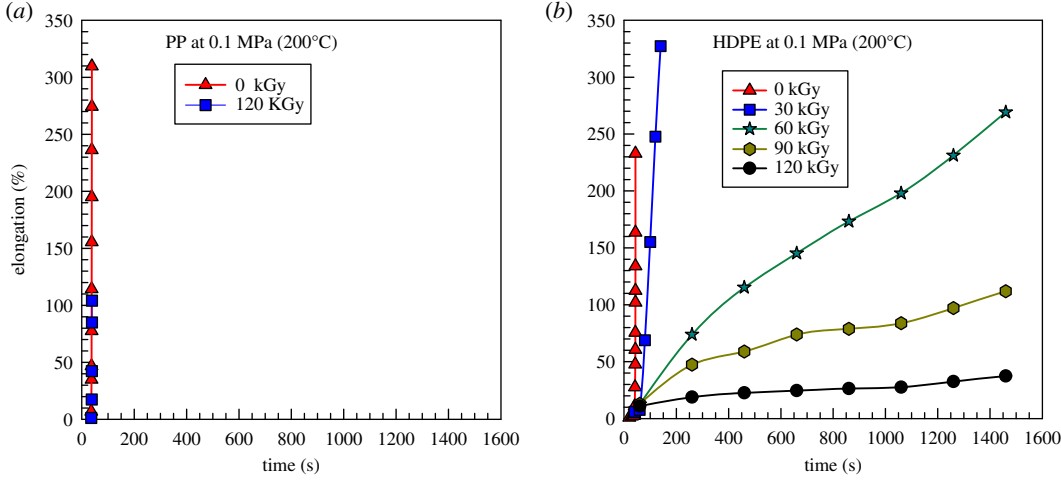

**Figure 1.** Plot of elongation versus time for (*a*) PP and (*b*) HDPE at 200°C and stress of 0.1 MPa for various irradiation doses.

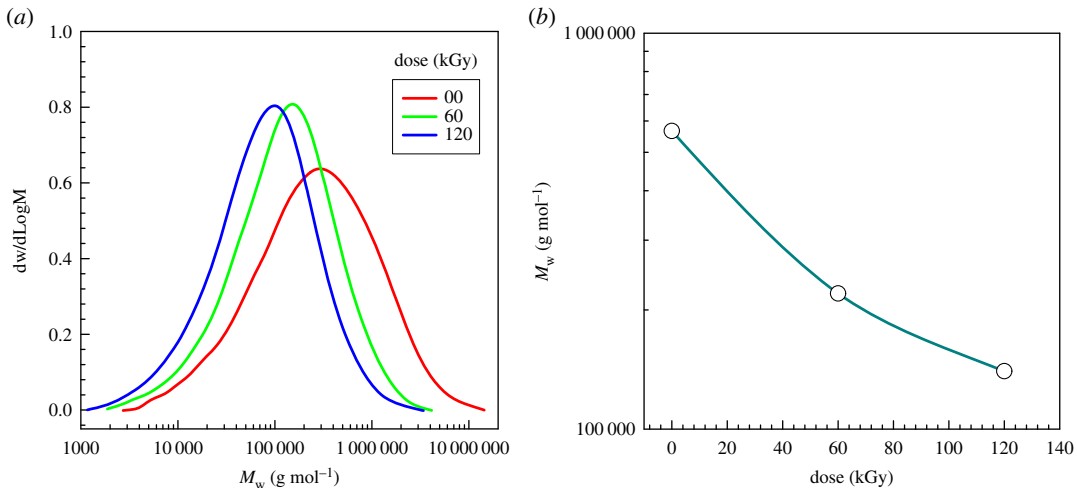

**Figure 2.** (*a,b*) Molecular weight distribution of PP (from GPC).

**Table 2.** Molecular weight of polypropylene by GPC.

| sample name | Mp | Mn | Mw | Mz | Mz+1 | PD |
|---|---|---|---|---|---|---|
| | g mol$^{-1}$ | | | | | |
| PP-00 kGy | 325 000 | 85 000 | 566 000 | 2 117 000 | 4 563 000 | 6.7 |
| PP-60 kGy | 155 000 | 51 000 | 220 000 | 610 000 | 1 180 000 | 4.3 |
| PP-120 kGy | 105 000 | 34 000 | 140 000 | 419 000 | 915 000 | 4.1 |

Before coming to the crystallization kinetics study, we have investigated mechanical properties in a wide temperature range (−150 to 200°C) with the help of dynamic mechanical analysis (DMA). Storage modulus (figure 3) of these two materials is very similar up to about 100°C, then HDPE loses its mechanical properties at about 130°C; for PP, this transition is located at about 150°C. At temperatures below the melting point, there was not a significant difference in modulus in pure samples versus the irradiated ones. There is only a small increase in storage modulus for HDPE in the temperature range 60–120°C (figure 3*b*) and a small decrease in tan delta (figure 3*c*). The cross-linking was done to a very small extent (compared to sulfur cross-linking in the rubber industry). Therefore, the change in storage modulus (up) and tan delta (down) is only very moderate. The creep test above the melting point (shown in figure 1*b*) is much more sensitive to such a small level of cross-linking.

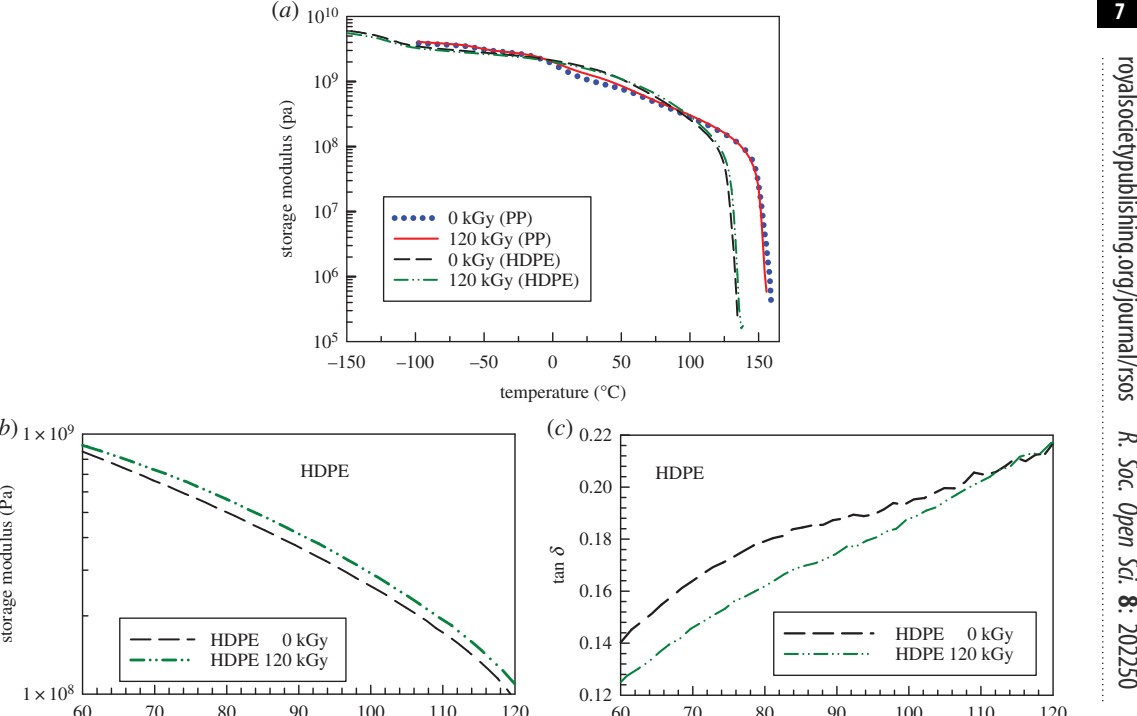

**Figure 3.** (*a–c*) Storage modulus and tan delta curves from DMA as a function of temperature for PP and HDPE.

FTIR is quite a powerful technique in the detection of chemical changes that might take place in these polymers during irradiation. While we have found almost no changes in the FTIR spectrum for HDPE (figure 4*b*), there was a small but detectable change of PP (figure 4*a*) in the 1500–1800 cm$^{-1}$ range. The range 1600–1700 cm$^{-1}$ is usually connected with a C=C bond and the area 1700–1770 cm$^{-1}$ with C=O in aldehydes, ketones or in carboxylic acids. It is perceivable that chain scission leads to a C=C formation at the end of a chain and that a small amount of oxygen could react with macro-radical rendering a C=O bond. Bearing in mind, the results from the high-temperature creep, GPC, DMA and FTIR, we will now focus on the crystallization behaviour.

Results of this analysis are shown in figure 5. Both polymers exhibit a decrease in bulk crystallization kinetics with the increasing irradiation level. It is clear that HDPE is much more affected than PP; in the case of HDPE, the $\tau_{1/2}^{-1}$ dropped from about 0.52 to about 0.03 min$^{-1}$, while in the case of PP, the decrease was much more moderate (from 0.3 to 0.13 min$^{-1}$); values for 0 and 120 kGy were compared. Apparently, the mobility (or diffusion rate) of macromolecules towards the crystallizing front of the lamella is considerably slowed down after cross-linking. Some of the cross-linked molecules could be completely prevented from any participation in the crystalline phase. This can be deduced from the crystallinity versus irradiation plot (figure 6). In contrast, the crystallinity of PP was not changed by irradiation.

The decrease in bulk crystallization rate with increasing molecular weight (in our case PE) was reported by many researchers [20,21,26–31] who explained this phenomenon by a higher mobility of the shorter molecules coming towards the lamella's growing front. The decrease in bulk crystallization rate with decreasing molecular weight (in our case PP) was also reported by Pospisil & Rybnikar [32] who investigated PPs with controlled rheology with various melt flow index and also by Ergoz *et al.* [33] in the Mn range about 4 k–20 k. Ou-Yang *et al.* [34] explored bulk crystallization kinetics of poly(ε-caprolactone) with Mn range 2.7 k–64.7 k g mol$^{-1}$. They found a maximum in the crystallization rate as a function of Mn. They concluded that increasing Mw has two opposite effects on crystallization. First, increasing Mw is reducing segmental mobility that leads to a lower crystallization rate. Second, increasing Mw increases the position of $T_m^0$ which increases the level of supercooling $\Delta T$ which increases the growth rate. Interplay between these two opposite effects can lead to the crystallization rate maximum as a function of Mw.

The bulk crystallization measured by DSC was also analysed by the Avrami equation (3.1), figure 7 and table 3. There is a tremendous difference in the crystallization kinetics expressed as a *k* parameter for

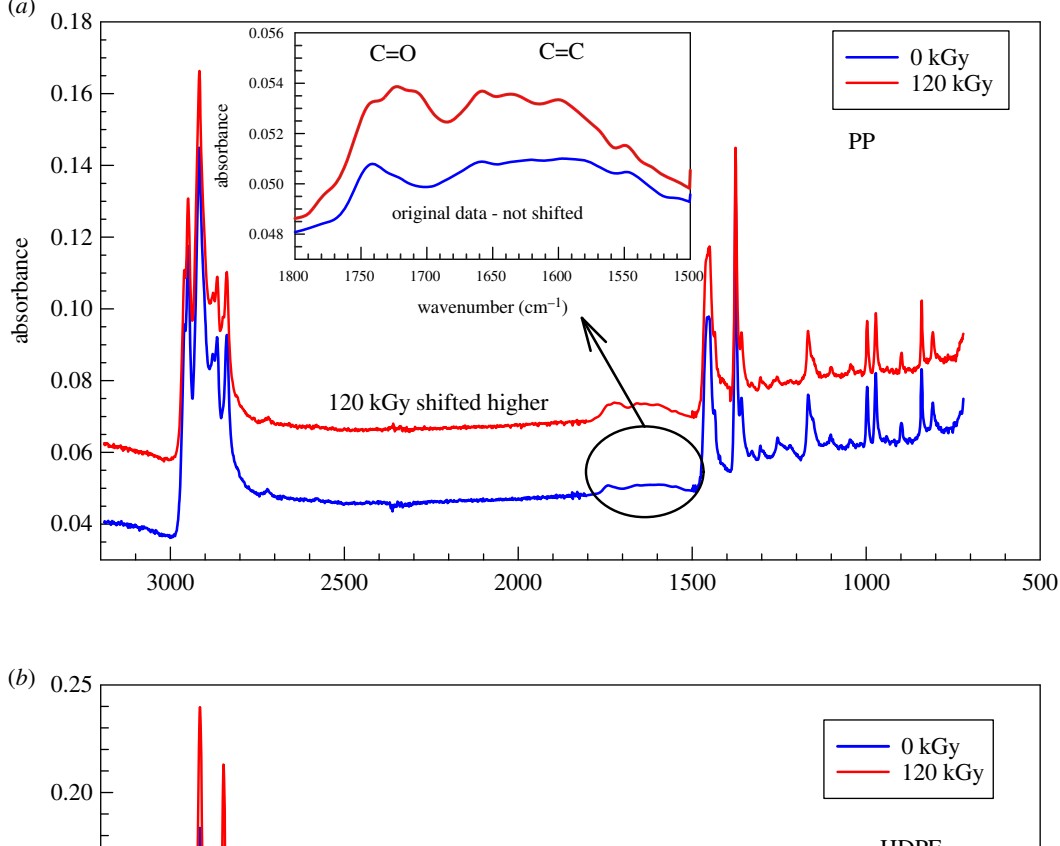

**Figure 4.** FTIR spectrum for (*a*) PP, and (*b*) HDPE with: 0 kGy and 120 kGy.

HDPE. In the case of PP, initially there is a notable decrease in the range from 0 to 60 kGy, but then the decrease in kinetics is much smaller; these results correspond well with the ones shown in figure 5. An additional parameter obtained from the Avrami analysis is *n*. There is quite a difference between HDPE and PP (2.2 versus 2.7), but not any significant difference between pure and irradiated samples. The *n* parameter is usually connected with two- or three-dimensional growth. Apparently, irradiation does not influence the number of dimensions in which the crystals can grow. As can be seen in table 3, PP was crystallized at 127°C. PP also crystallized at other temperatures (121–131°C), and it was evaluated by the Avrami equation. The results are presented in table 4. The *k* parameter is always decreasing with the increasing crystallization temperature. The *n* parameters for $PP_{0kGy}$ have values of around 2.2, while for $PP_{120kGy}$ the *n* values are slightly higher (range 2.52–2.84).

While figure 5 compares the crystallization kinetics at a fixed temperature for different irradiation levels, figure 8 shows the dependence of the crystallization kinetics on temperature only for pure PP and $PP_{120kGy}$. Again, the $PP_{120kGy}$ crystallization is slower for all evaluated temperatures and the kinetics increases exponentially with the decreasing temperature. DSC is a very powerful technique

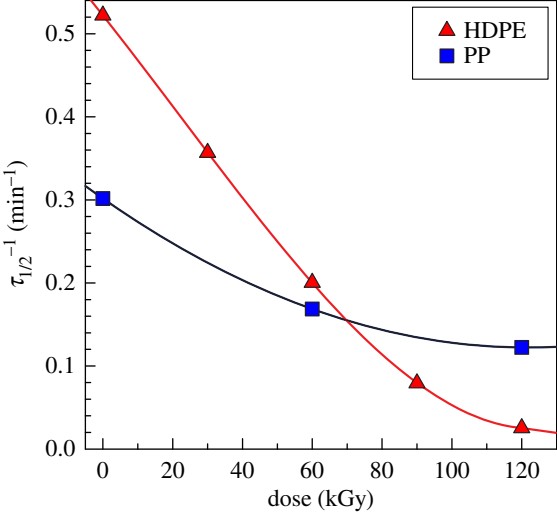

**Figure 5.** Crystallization kinetics versus irradiation dose (kGy) for PP ($T_c = 127°C$) and HDPE ($T_c = 122°C$) from DSC.

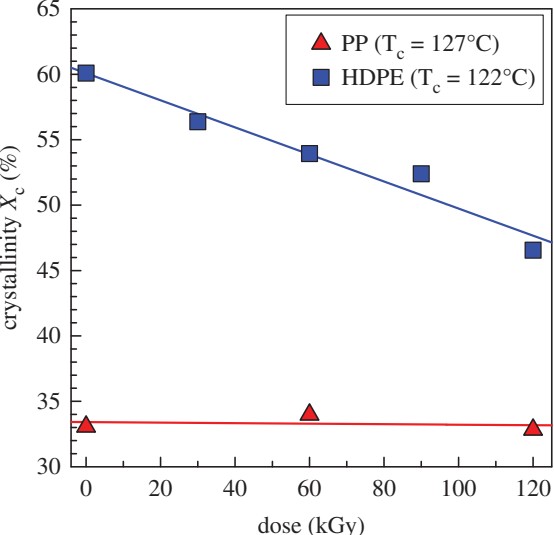

**Figure 6.** Crystallinity ($X_c$) (after second heating) versus irradiation dose (kGy) for PP ($T_c = 127°C$) and HDPE ($T_c = 122°C$) from DSC.

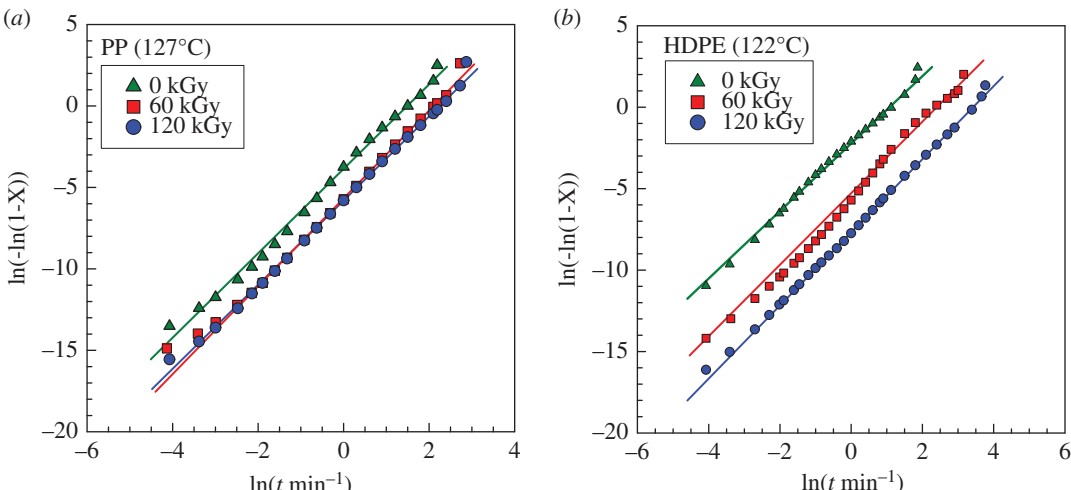

**Figure 7.** Avrami plots for (*a*) PP and (*b*) HDPE at various irradiation doses from DSC.

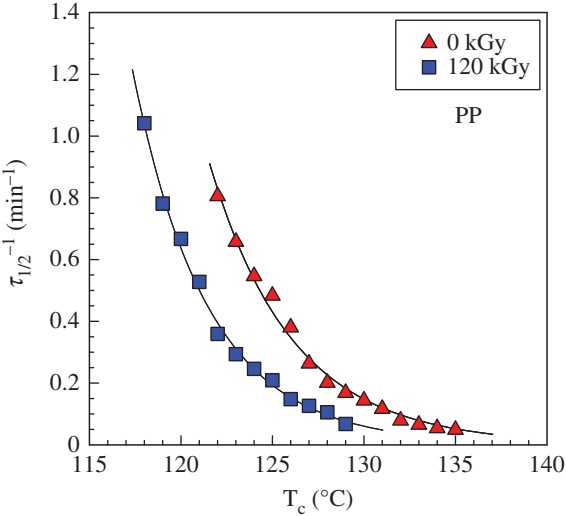

**Figure 8.** Crystallization temperatures versus crystallization kinetics plot for PP from DSC.

**Table 3.** Avrami parameters from DSC.

| samples | $T_c$ (°C) | 0 kGy | | 60 kGy | | 120 kGy | |
|---|---|---|---|---|---|---|---|
| | | $n$ | $k$ (m$^{-1}$) | $n$ | $k$ (m$^{-1}$) | $n$ | $k$ (m$^{-1}$) |
| PP | 127 | 2.59 | 0.0211 | 2.70 | 0.0034 | 2.58 | 0.0030 |
| HDPE | 122 | 2.07 | 0.1070 | 2.20 | 0.0050 | 2.24 | 0.0004 |

**Table 4.** Avrami parameters for PP from DSC.

| radiation dose (kGy) | crystallization temperature (°C) | | | | | | | | | |
|---|---|---|---|---|---|---|---|---|---|---|
| | 121 | | 123 | | 125 | | 129 | | 131 | |
| | $N$ | $k$ (m$^{-1}$) | $n$ | $k$ (m$^{-1}$) | $n$ | $k$ (m$^{-1}$) | $n$ | $k$ (m$^{-1}$) | $n$ | $k$ (m$^{-1}$) |
| 0 | — | — | — | — | 2.27 | 0.1076 | 2.34 | 0.0089 | 2.16 | 0.0039 |
| 120 | 2.52 | 0.1065 | 2.68 | 0.0217 | 2.84 | 0.0053 | — | — | — | — |

for the evaluation of bulk crystallization kinetics. However, it is not clear from the DSC measurement if the decrease in crystallization kinetics comes from a smaller number of nucleation centres or from slowly growing spherulites or perhaps from both. The hot-stage optical microscopy can clarify this issue.

Figure 9 illustrates the growth of spherulites as observed by optical microscopy at 140°C. It is clear that both pure PP and the sample irradiated by 30 kGy have a higher number of spherulites in the observed area than the samples irradiated by 60, 90 and 120 kGy. The increased level of irradiation caused a decrease in the number of nucleation centres. This result agrees well with the bulk crystallization kinetics decrease observed by DSC that was shown in figure 5. For pure PP and PP irradiated by 30 kGy, space is filled very quickly with small spherulites that truncate into each other and then the crystallization stops. Most likely the macromolecules in pure PP do not move very far from their original positions during melting. The memory of the original chain positions remains in the material and it makes the nucleation step easier [35–38]. On the other hand, the irradiation 60–120 kGy has apparently caused some change in the chemical structure of some macromolecules. The chain scission or branching influenced the ability of chains to fold into a lamellar structure

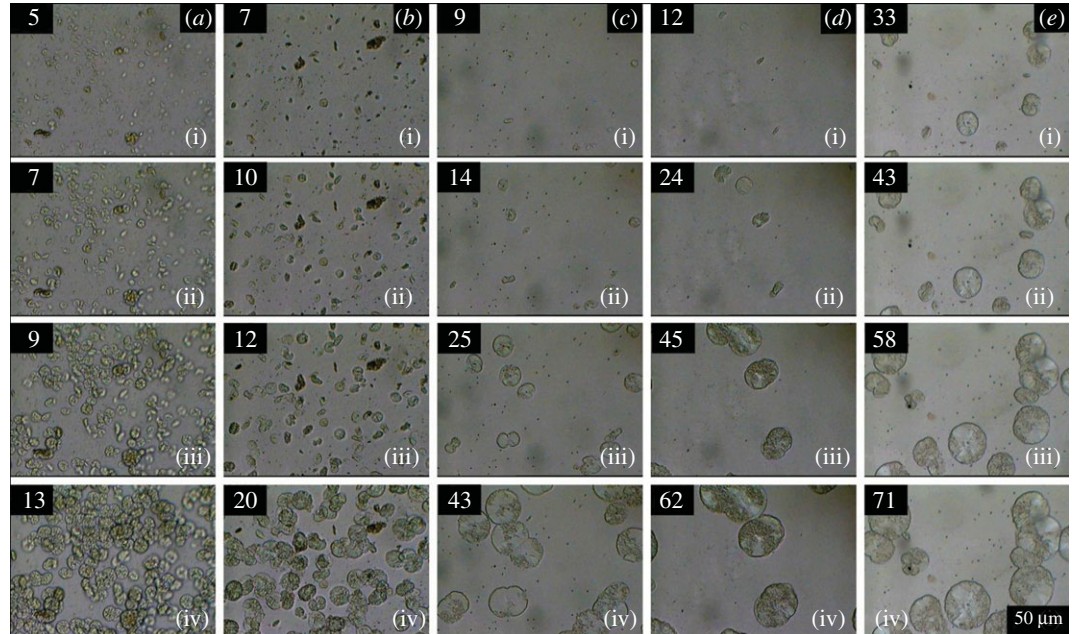

**Figure 9.** Growth of spherulites in time at $T_c = 140°C$ for various irradiation doses: (*a*) 0 kGy, (*b*) 30 kGy, (*c*) 60 kGy, (*d*) 90 kGy, and (*e*) 120 kGy for PP by hot-stage optical microscopy after 1 min pre-heating at 200°C (numbers in upper left corner are time of crystallization in min).

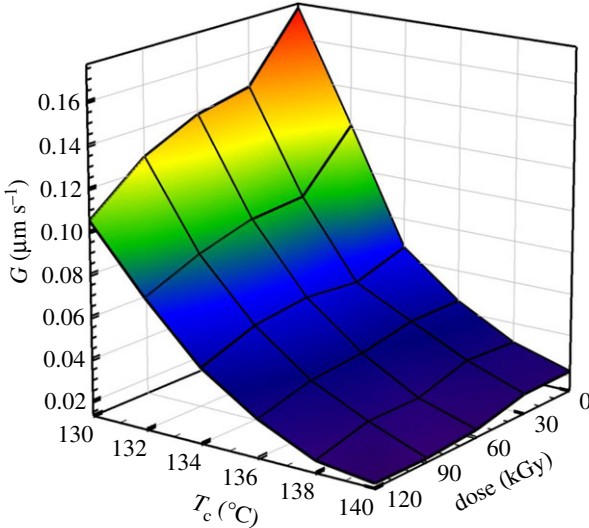

**Figure 10.** Spherulites growth rate as a function of crystallization temperature ($T_c$) and irradiation dose for PP from optical microscopy.

[21,27,29,32,33]. This is manifested by a much lower number of nucleation centres and also by a slower radial growth of the individual spherulites. The decrease in the growth rate of individual spherulites is quantitatively illustrated in figure 10. At all crystallization temperatures, the growth rate gradually decreased with the increasing radiation level.

Various results were found in the literature for polarized optical microscopy measurements. Some researchers reported a decrease in the crystallization rate with increasing Mw [21,28]. However, there is a report showing the existence of a maximum [39] and reports showing no change at all [32,40,41].

The crystallization kinetics of individual spherulites was evaluated in the temperature range of 130–140°C. An example of this analysis is shown in figure 11 for PP irradiated to 90 kGy. While at 130°C it took only 7 min to grow about 100 μm spherulites, at 140°C the same thing took about an hour. Hence the influence of the crystallization temperature is very significant. The strong influence of

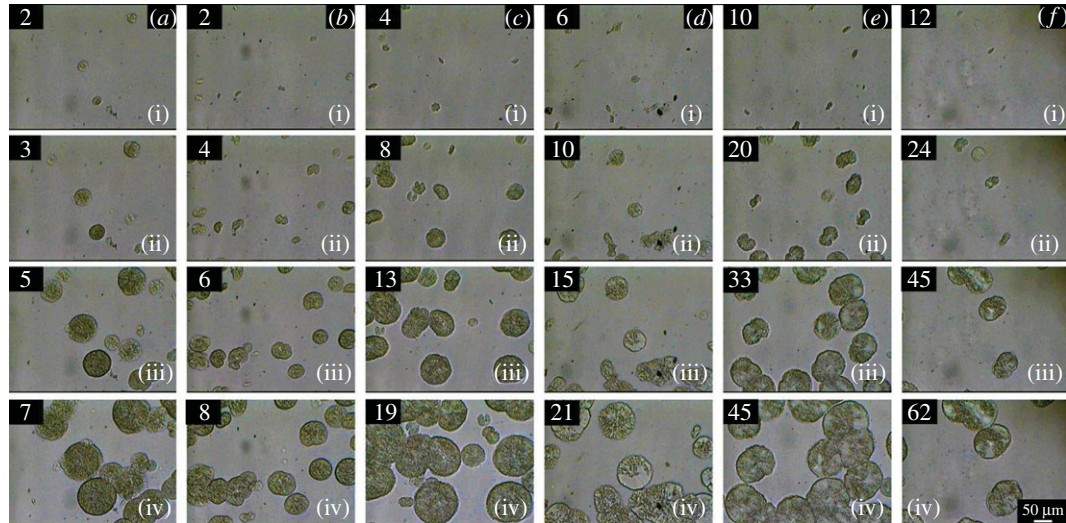

**Figure 11.** Growth of spherulites in time at 90 kGy at various crystallization temperatures: (*a*) 130°C, (*b*) 132°C, (*c*) 134°C, (*d*) 136°C, (*e*) 138°C, and (*f*) 140°C for PP by hot-stage optical microscopy after 1 min pre-heating at 200°C (numbers in upper left corner are time of crystallization in min).

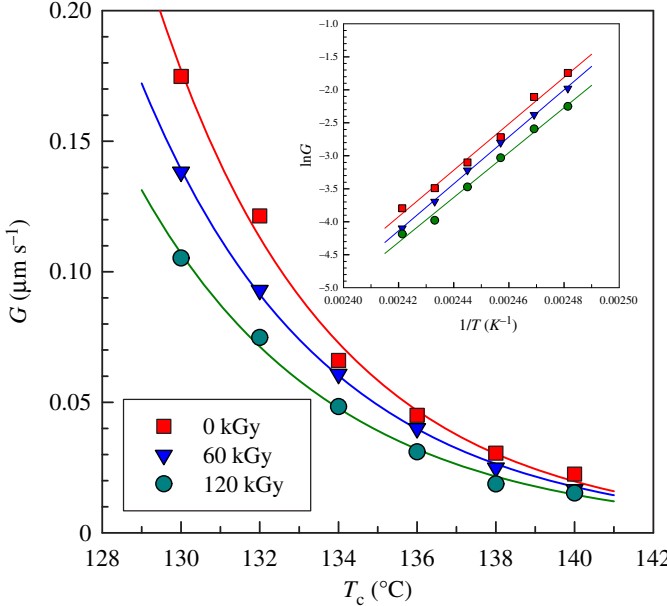

**Figure 12.** Spherulites growth rate (*G*) versus crystallization temperatures for PP at various irradiation doses from optical microscopy.

crystallization temperature is nicely visible in figure 12 where the dependence of $G$ on $T_c$ fitted best by the Arrhenius equation (equation (3.6)).

Initially, we performed the analysis by linear regression of the Arrhenius equation in the logarithm form, equation (3.7) which gave us the estimated parameters $E_a$ and $A$. This linear regression is shown in the inset of figure 12. Then these estimated parameters $E_a$ and $A$ were used for a much more precise nonlinear regression analysis shown in figure 12. This nonlinear analysis has revealed a decrease in activation energy with the increasing radiation level (table 5) which is mainly caused by a considerable decrease in crystallization kinetics of $PP_{60kGy}$ and $PP_{120kGy}$ samples at lower crystallization temperatures. Again there is a large difference in the activation energy between samples $PP_{0kGy}$ and $PP_{60kGy}$ compared to $PP_{60kGy}$ and $PP_{120kGy}$.

The Hoffman–Lauritzen analysis of crystallization was performed for PP samples measured by optical microscopy (figure 13). This analysis revealed two crystallization regimes, II and III with the transition being around 136–138°C. Table 5 summarizes the results of the Hoffman-Lauritzen analysis.

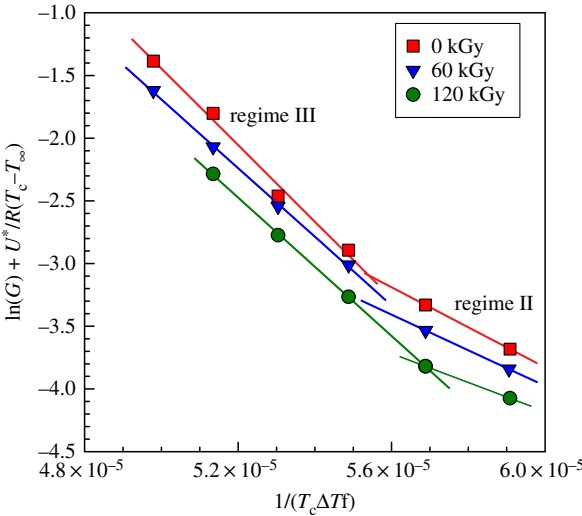

**Figure 13.** Hoffman–Lauritzen plots for PP at various irradiation doses from optical microscopy.

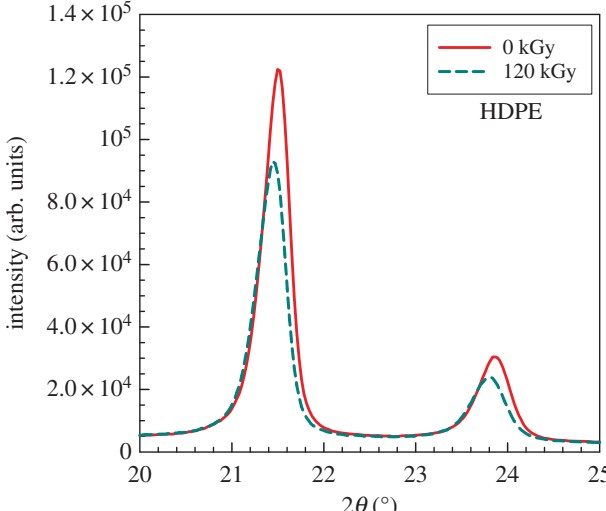

**Figure 14.** XRD analysis for HDPE.

**Table 5.** Hoffman–Lauritzen parameters and activation energy of PP by Arrhenius plot from optical analysis.

| radiation dose (kGy) | regime III | | regime II | | |
|---|---|---|---|---|---|
| | $K_g \times 10^{-5}$ (K²) | $\ln G_0$ | $K_g \times 10^{-5}$ (K²) | $\ln G_0$ | $E_a$ (kJ mol⁻¹) |
| 0 | 3.05 | 13.83 | 1.60 | 5.75 | 303 |
| 60 | 2.73 | 12.00 | 1.42 | 4.55 | 286 |
| 120 | 2.76 | 11.87 | 1.15 | 2.75 | 275 |

Again the values of $\ln G_0$ (corresponding to kinetics) were decreasing with the irradiation dose, while the $K_g$ parameters were not influenced much.

In addition to the crystallization kinetics analysis performed by DSC and by optical microscopy, we have analysed the crystalline structure also by X-ray diffraction (XRD) analysis. Figure 14 shows that irradiation has caused a decrease in the intensity of two XRD peaks for HDPE which is in good agreement with their lower crystallinity observed by DSC after second melting (figure 6). Not only does the peak intensity decrease but also apparently the peaks shift to lower angles. This

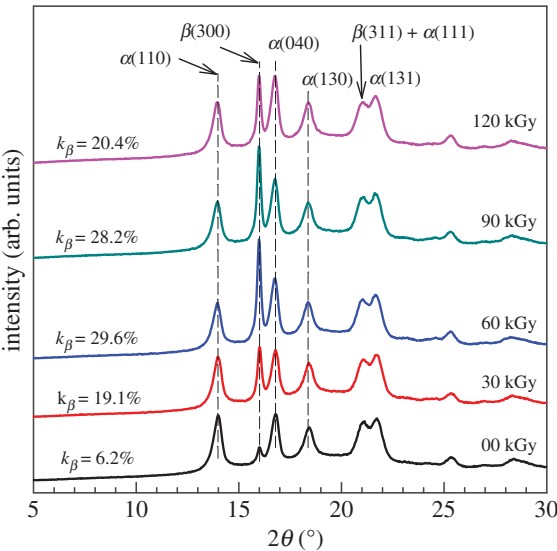

**Figure 15.** Wide angle XRD analysis for PP.

phenomenon can be explained by the destruction of smaller crystals by radiation (larger crystals remain). The situation was quite different in the case of PP (figure 15). The large peaks did not change almost at all; this observation corresponds well with the unchanged crystallinity shown in figure 6. However, we found an interesting increase in the peak for $\beta$-phase. Apparently, irradiation helps to generate $\beta$-phase up to about 60 kGy, then the intensity at $2\theta = 16°$ starts to decrease again (figure 15).

Jones *et al*. evaluated the relative proportion of $\alpha$ and $\beta$ forms by an empirical ratio $k$ [42] where

$$k = \frac{H\beta_1}{H\beta_1 + (H\alpha_1 + H\alpha_2 + H\alpha_3)}, \tag{4.1}$$

and $H\alpha_1$, $H\alpha_2$ and $H\alpha_3$ are the heights of three strong equatorial $\alpha$-form peaks (110), (040) and (130), above the background curve, and $H\beta_1$ the height of the strong single (hk0) $\beta$ peak at $d = 5.495$ Å.

Lu *et al*. have initially calculated the total crystalline fraction, $X_c$, by the following equation [43]:

$$X_c = \frac{\sum A_{\mathrm{cryst}}}{\sum A_{\mathrm{cryst}} + \sum A_{\mathrm{amorp}}}, \tag{4.2}$$

where $A_{\mathrm{cryst}}$ and $A_{\mathrm{amorp}}$ are the fitted areas of crystal and amorphous region of wide angle XRD (WAXD) curves, respectively. Then they calculated the content of $\beta$-phase, $X_\beta$, according to this equation [43]:

$$X_\beta = \frac{A_{\beta(300)}}{A_{\beta(300)} + A_{\alpha(110)} + A_{\alpha(040)} + A_{\alpha(130)}} \times X_c, \tag{4.3}$$

where $A_{\beta(300)}$, $A_{\alpha(110)}$, $A_{\alpha(040)}$, and $A_{\alpha(130)}$ represent the reflection peak areas of the corresponding crystallographic planes. The $\beta$-phase could be achieved by increased shear stress by crystallization with temperature gradient or by the addition of $\beta$ nucleating agents [43].

Apparently, according to our results, the e-beam radiation can promote the formation of $\beta$ phase too. To look at the data in detail, we have calculated the total crystalline fraction $X_c$ according to equation (4.2) and also evaluated the content of $\beta$-phase $X_\beta$ according to equation (4.3). The content of $\alpha$-phase was evaluated by

$$X_\alpha = X_c - X_\beta. \tag{4.4}$$

The results are plotted in figure 16. It is interesting that the total crystallinity $X_c$ initially increases in the range 0–60 kGy, then it levels off. In the same radiation range, there is a significant increase in $\beta$-phase and, at the same time, a decrease in $\alpha$-phase. The decrease in chain length and formation of new groups (C=C, C=O…) has slightly damaged the initially formed $\alpha$-crystals, probably on the surface of the lamellae. The shorter chains were able to regroup and form the $\beta$-crystals. In the 60–120 kGy irradiation range, the content of $\alpha$-crystals increases and the content of $\beta$-crystals decreases. This could be connected to a rather lower stability of $\beta$-phase that is susceptible to transition to $\alpha$-phase during further irradiation.

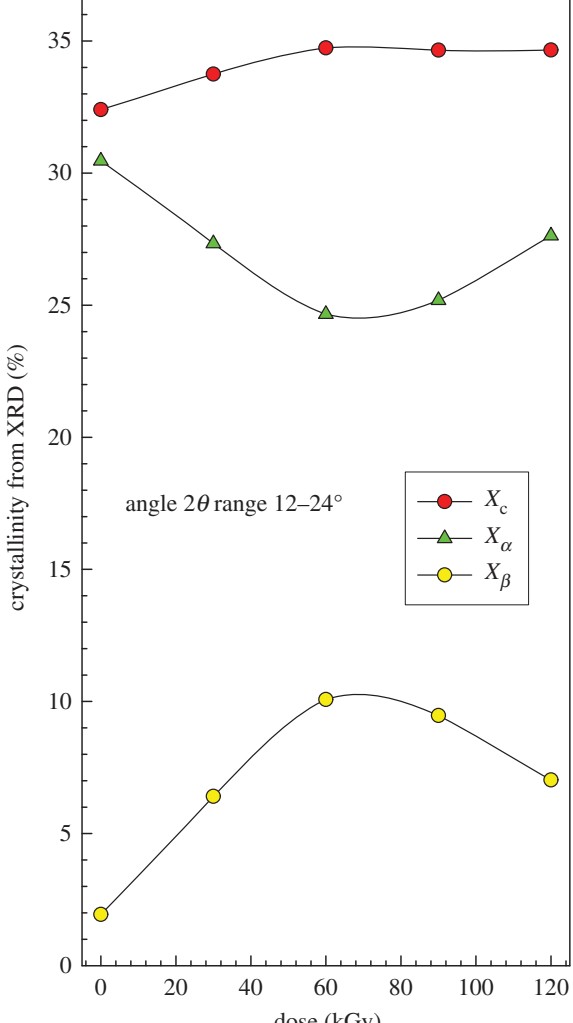

**Figure 16.** WAXD analysis for PP–continuation.

Irradiation could cause a generation of disorders of the crystalline lattice of the lamellae of the $\alpha$-phase. We have observed a significant decrease in the melting point (figure 17a, first scan in DSC) which resembles a decrease in the melting point owing to the decrease in lamellar thickness. The whole lamella cannot have decreased thickness but only local defects exist. Damaged spots of the lamellae melt during heating at a lower temperature. The change of the melting point with lamella thickness is well described by the Gibbs–Thomson equation. When we assume a lamellar crystal with lateral sizes $a$ and $b$ and thickness $l$, the melting temperature ($T_m$) of lamellar crystal with $l(T_m(l))$ is given by [44]:

$$T_m(l) = T_m^0 \left(1 - \frac{2}{\Delta h}\left(\frac{\sigma}{a} + \frac{\sigma}{b} + \frac{\sigma_e}{l}\right)\right), \tag{4.5}$$

where $\sigma$ is the surface free energy, $\sigma_e$ is the end surface free energy and $\Delta h$ is the heat of fusion. Generally, $a$ and $b$ are much larger than $l$ in the case of a lamellar crystal. Therefore, equation (4.5) can be approximated by

$$T_m(l) = T_m^0 - \frac{C}{l}, \;\; \text{where} \;\; C = \frac{2\sigma_e T_m^0}{\Delta h}. \tag{4.6}$$

For one material, one could estimate the decrease in lamellar thickness from the melting point depression:

$$\frac{l_2}{l_1} = \frac{T_m^0 - T_{m1}(l)}{T_m^0 - T_{m2}(l)}. \tag{4.7}$$

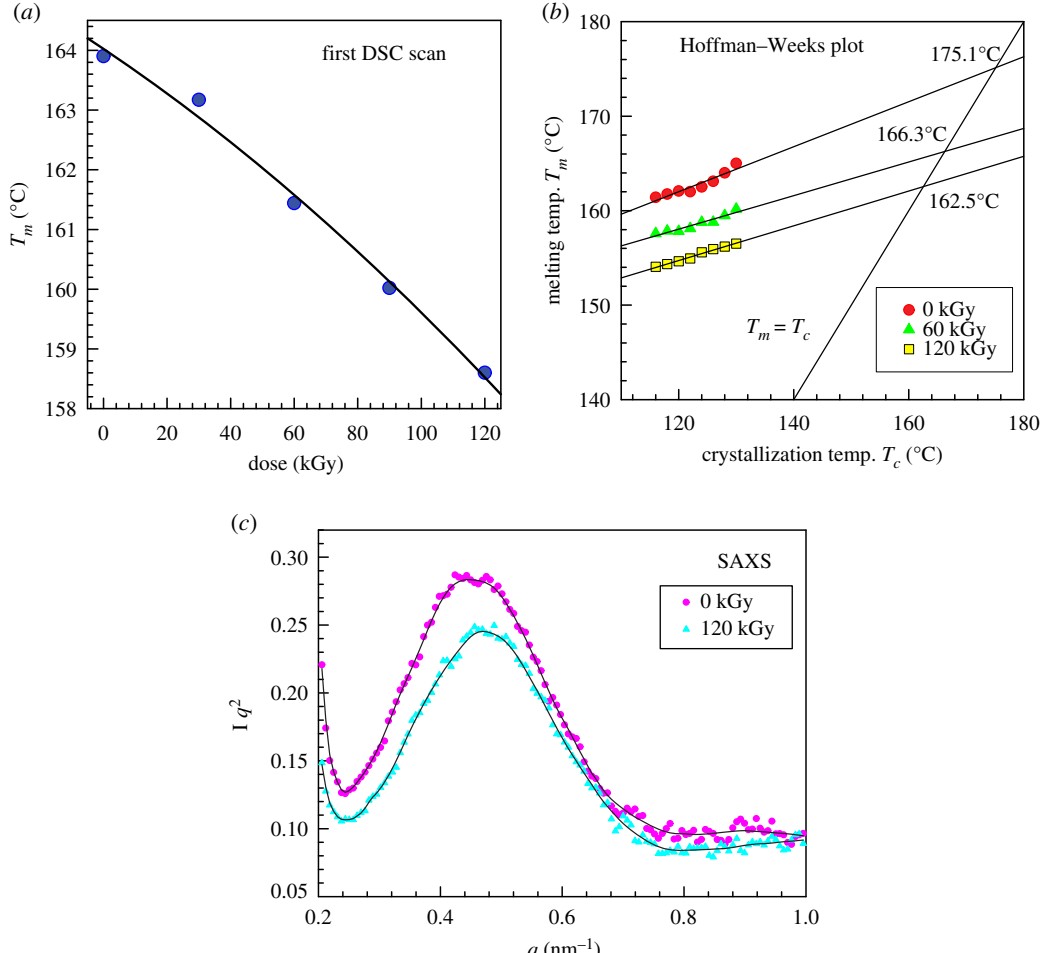

**Figure 17.** PP results: (a) first DSC scan, (b) Hoffman–Weeks plot to determine $T_m^0$, (c) Lorentz-corrected 1D-SAXS intensity profiles.

**Table 6.** DSC results for PP: melting points from first and second heating scan ($T_m$), decrease in melting point owing to irradiation $\Delta T_m$, equilibrium melting points from Hoffman–Weeks plot $T_m^0$.

| dose | first scan | first scan | second scan | second scan | |
|---|---|---|---|---|---|
| | $T_m$ | $\Delta T_m$ | $T_m$ | $\Delta T_m$ | $T_m^0$ |
| (kGy) | (°C) | (°C) | (°C) | (°C) | (°C) |
| 0 | 163.90 | 0.00 | 161.41 | 0.00 | 175.13 |
| 60 | 161.44 | 2.46 | 157.59 | 3.82 | 166.27 |
| 120 | 158.60 | 5.30 | 154.06 | 7.35 | 162.55 |

For the $l_2/l_1$ calculation, the equilibrium melting point value is necessary. It can be obtained by a Hoffman–Weeks plot. This is shown in figure 17b. After irradiation, the $T_m^0$ decreases. One has to be careful, though, because during the Hoffman–Weeks measurement the sample is melted, crystallized and then melted again, and in the case of irradiated PP, the chains are shorter and there are chemical defects in the form of oxygen-containing groups. Thus, during the second melting, we also get a different melting point and the melting point depression is greater than that during the first DSC scan (table 6). If we assumed the existence of only one $T_m^0$ (before any melting or recrystallization to make the Hoffman–Weeks plot), we could estimate the apparent decrease in lamellar thickness after 60 kGy irradiation according to equation (4.7): $l_2/l_1 = (175.13 - 163.90)/(175.13 - 161.44) = 0.8203$.

In figure 4, we have shown by FTIR the presence of C=O groups in irradiated PP. It is conceivable that there is a large content of oxygen in the amorphous phase compared to the crystalline phase. A released

chain from the surface of the lamella (tie molecule) is able to crystallize further in the amorphous region, this time in β-form. The detailed analysis of WAXD results is presented in table 7 and in figure 16. The Bragg's diffraction law was used in the results evaluation:

$$n\lambda = 2d\sin\theta \quad n = 1, 2, 3, \ldots \tag{4.8}$$

This basic equation shows that for a given value of the x-ray wavelength $\lambda$, measurement of the angle $\theta$ gives the information on the spacing between planes through the scattering centres that make up the crystal.

The crystallite size of each plane can be calculated from diffraction peaks of WAXD using the Scherrer equation [45,46]:

$$L_{hkl} = \frac{K\lambda}{\beta_{hkl}\cos\theta_{hkl}}, \tag{4.9}$$

where $L_{hkl}$ is the crystallite diameter ($hkl$), $\lambda$ is the wavelength of the X-ray, $K$ is the crystallite shape factor, $\theta_{hkl}$ is the Bragg angle and $\beta_{hkl}$ is the full width of the direction line at half-maximum intensity measured in radians.

From the results, it is clear that by irradiation, β-phase increases steadily and α-phase decreases steadily. However, at 60 kGy, the unstable β-phase starts to change to a more stable α-phase. Oxygen at the chain end after scission cannot enter the compact lamellae and crystallization requires a longer time. Shorter chains have a higher thermal movement and therefore they require a higher degree of supercooling to assemble into nucleation centres. Thus crystallization happens at lower temperatures (lower $T_c$). At the crystallization temperature, shorter chains have higher difficulty to attach towards the lamella edge because the process is more intensively disturbed by a higher level of thermal movement.

The influence of e-beam radiation on the molecular and chemical structure of some parts of PP molecules is as follows. A consequence of the influence of e-beam radiation on the PP is generally a known chain scission in the presence of $O_2$ by the appropriate chemical mechanism [4] when hydroperoxides/dialkyl peroxides, tertiary alcohol and methyl ketone are generated (scheme 1).

In the crystalline phase, there is an increase in the number of built-in chain ends and in chemically bonded oxygen-containing groups which generally represent a defect in the crystalline lattice. These groups in the crystalline plane cause a reduction in XRD reflection at the appropriate angle, i.e. reduction of the intensity of a studied diffraction peak. Defects in the crystalline lattice are irreversible and together with molecular weight decrease (chain scission) they participate in the reduction of mechanical properties and disappear after melting. They can no longer get into the newly created crystalline lattice because the regular chain structure is disrupted, and thus they remain only in the amorphous phase.

In the amorphous phase of the PP, the e-beam radiation causes the same changes at the chain level, but their manifestation is somewhat different. The main phenomenon is the chain scission (reduction in the molecular weight) with all the resulting consequences. This is manifested especially after subsequent melting as a reduction in the viscosity of the melt, which can significantly affect the progress of the second crystallization.

In the amorphous phase, however, according to the measured XRD (figures 15 and 16) and DSC results (figure 17a), still another phenomenon occurs during irradiation. In the amorphous phase, preferably a new crystalline β-phase is generated, which subsequently converts into α-phase. A possible explanation of this phenomenon is the release of a part of the chains after scission from the state of partial incorporation in the crystalline phase (tie molecules). Once released, they gain such great mobility that they can create together another part, this time β-phase. As they are thermodynamically less stable, they can change by the influence of high-energy irradiation into the α-phase. Surprisingly, we observed the subsequent increase in α-phase during the irradiation process (figure 16, curve $X_\alpha$). However, this does not alter the fact that after irradiation the main crystalline α-phase contains defects in its crystalline lattice. Owing to the fact that the irradiation of the samples proceeded from one direction, the frequency of the defects can vary in various crystallization planes (110), (040) and (130), and indeed this has also been observed (table 7).

We have investigated the crystalline structure also with the help of SAXS (figure 17c and table 8). The long period was calculated using equation (4.10) (Bragg's law) [45]:

$$L = \frac{2\pi}{q_{max}}. \tag{4.10}$$

**Table 7.** Parameters from WAXD analysis of PP; d-spacing is from Bragg's equation and $L_{hkl}$ is from Scherrer equation, FWHM is the full width at half-maximum.

| dose (kGy) | area | area (%) | FWHM | centre | height | d-spacing (Å) | $L_{hkl}$ (nm) |
|---|---|---|---|---|---|---|---|
| **peak 1 - alpha (110)** | | | | | | | |
| 0 | 5618 | 41.23 | 0.4765 | 13.98 | 9742 | 6.326 | 17.94 |
| 30 | 4978 | 33.41 | 0.4672 | 13.98 | 8804 | 6.326 | 18.30 |
| 60 | 4364 | 24.79 | 0.4660 | 13.96 | 7824 | 6.338 | 18.34 |
| 90 | 4342 | 25.52 | 0.4673 | 13.96 | 7866 | 6.338 | 18.29 |
| 120 | 4727 | 27.09 | 0.4553 | 13.96 | 8671 | 6.338 | 18.77 |
| **peak 2 - beta (300)** | | | | | | | |
| 0 | 844 | 6.20 | 0.2736 | 16.01 | 2687 | 5.529 | 31.54 |
| 30 | 2847 | 19.10 | 0.2544 | 16.01 | 9223 | 5.529 | 33.92 |
| 60 | 5207 | 29.57 | 0.2374 | 15.99 | 17602 | 5.538 | 36.34 |
| 90 | 4802 | 28.22 | 0.2388 | 15.99 | 16311 | 5.538 | 36.12 |
| 120 | 3554 | 20.37 | 0.2400 | 15.99 | 11913 | 5.538 | 35.95 |

(*Continued.*)

**Table 7.** (*Continued.*)

| dose (kGy) | area | area (%) | FWHM | centre | height | d-spacing (Å) | $L_{hkl}$ (nm) |
|---|---|---|---|---|---|---|---|
| peak 3 - alpha (040) | | | | | | | |
| 0 | 3965 | 29.10 | 0.4275 | 16.79 | 8458 | 5.274 | 20.26 |
| 30 | 3905 | 26.21 | 0.4243 | 16.79 | 8436 | 5.274 | 20.42 |
| 60 | 5000 | 28.40 | 0.4249 | 16.77 | 10737 | 5.282 | 20.39 |
| 90 | 4810 | 28.27 | 0.4243 | 16.77 | 10407 | 5.282 | 20.42 |
| 120 | 5508 | 31.57 | 0.4195 | 16.77 | 11844 | 5.282 | 20.65 |
| peak 4 - alpha (130) | | | | | | | |
| 0 | 3197 | 23.47 | 0.5060 | 18.43 | 5320 | 4.809 | 17.28 |
| 30 | 3171 | 21.28 | 0.4966 | 18.40 | 5394 | 4.815 | 17.60 |
| 60 | 3036 | 17.24 | 0.4928 | 18.38 | 5457 | 4.822 | 17.74 |
| 90 | 3061 | 17.99 | 0.4957 | 18.38 | 5362 | 4.822 | 17.63 |
| 120 | 3657 | 20.96 | 0.4925 | 18.40 | 6232 | 4.815 | 17.75 |

**Scheme 1.** Radiation oxidation mechanism of PP.

**Table 8.** SAXS results for PP: long period $L$ was calculated from Bragg's law.

| dose (kGy) | centre nm$^{-1}$ | height arb. units | FWHM nm$^{-1}$ | $L$ nm |
|---|---|---|---|---|
| 0 | 0.4584 | 0.1689 | 0.2228 | 13.71 |
| 120 | 0.4758 | 0.1508 | 0.2367 | 13.21 |

There is only a small decrease in the long period owing to irradiation. The original sample had $L = 13.7$ nm while the sample irradiated to 120 kGy had $L = 13.2$ nm.

The decrease in the chain length and generation of the defects in the crystalline lattice can thus be considered as the main reason for the reduction of the mechanical properties (more specifically elongation at break) of the irradiated PP as it is generally presented in the literature [4,47].

Nedkov *et al*. studied the effect of gamma irradiation on the PP. On the surface of the samples, they observed that reflex 300 increased with increasing doses of irradiation, i.e. the $\beta$-phase increased, whereas $\alpha$-phase decreased slightly. They first discussed radiation up to 100 kGy. In amorphous regions, a chain break prevailed. This resulted in an increase in the number of chain ends, as well as in pushing off the irradiation defects from the crystalline part towards the boundary layer of the lamellae. The latter phenomenon improved crystallite perfection. In the case of irradiation of 100–2000 kGy both regions (lamellar and also amorphous) were damaged. The average amorphous layer expanded from 4 to 12 nm. The energy stored in this area releases and could cause significant chemical transformations, whose final results are cross-linking, grafting and a great number of short chains. The disorder in the crystal part also increases and the crystal and amorphous densities have closer values. This process resembles thermal melting and because of that it was named partial radiation melting [48].

Manas *et al*. studied the influence of beta irradiation on the mechanical properties of PP. They wrote that when irradiation doses were high the structural changes were related to radiation melting. Some defects or mayhems in the crystal lamella were pushed to its surface. Most probably it was owing to the transport of energy by excitons along with the macromolecules and, in this way, the lamella surface became more defective, hence lamella thickness decreased a little. Destructions and cross-linking occurred simultaneously predominately in disordered zones [49].

Bhateja *et al*. described the molecular mechanism during irradiation. Common to all the observations was the scission of tie molecules by high-energy irradiation, assisted in some cases by the presence of oxygen and inhibited in others by the presence of radical scavengers. Tie molecule scission produces a number of effects. First, there is an immediate relief of stress in the crystal to which the tie molecule is attached. This gives rise to an instantaneous increase in melting temperature and heat of fusion without

any change in the physical dimensions of the crystal. Second, the scission of tie molecules also allows the development of new crystalline volume. In contrast with the strain-relief mechanism referred to above, this process cannot be instantaneous, because crystal growth requires time. For polymers with folded chain crystals, like melt-processed PEs, there is ample evidence for the growth of new lamellae and virtually none for lamellar thickening. Because the new lamellae normally grow at ambient temperatures, their growth rate is slow and their thickness small. This gives rise to very long-term ageing effects in PE and manifests itself in DSC traces by the appearance of low-melting crystalline fractions, which increase in amount with time. These long-term effects may be assisted by ongoing oxidative scission of molecules, but even this seems to be related to the initial radiation treatment. The underlying mechanism in the radiation-induced crystallization of polytetrafluoroethylene was shown schematically. This mechanism is similar to that proposed earlier for ultra high molecular weight PE in which the chain scission in the amorphous phase is followed by the growth of new lamellae in these regions [50].

## 5. Conclusion

The high-temperature (200°C) creep test confirmed that e-beam radiation causes cross-linking in HDPE and chain scission in PP. The decrease of PP's Mw was monitored by GPC. FTIR has pointed out the presence of C=C and C=O bonds in PP after irradiation. The crystallization kinetics study revealed a tremendous decrease in the cross-linked HDPE and a moderate decrease in PP. Also, while crystallinity remained unchanged by irradiation for PP, it was decreased significantly (from 60 to 47%) for HDPE. Optical microscopy clearly illustrated a smaller number of nucleation centres in PP after irradiation and also a decreased rate of crystallization of individual spherulites. XRD analysis exposed a lower crystallinity for HDPE and a very interesting increase of $\beta$-phase in PP with the maximum being at 60 kGy.

Our results of crystallization kinetics influenced by irradiation show a decrease in kinetics for both PE and PP, even though the causes are completely different because PP (in contrast with PE) contains a tertiary carbon atom in the main chain. In the case of PE, there is an increase in molecular weight, branching and cross-linking. In the case of PP, we found a decrease in the following: molecular weight, bulk crystallization kinetics, spherulitic crystallization kinetics and the number of spherulites. These results correspond well with other researchers [33] who demonstrated a 'U' shape of $\tau_{0.01}$ which is inversely proportional to crystallization kinetics as a function of Mw. In the lower molecular weight range, the decrease in Mw leads to higher $\tau_{0.01}$ (or slower crystallization). In a higher molecular weight range, the increase in Mw leads also to slower crystallization explained by the reptation model [29].

Explanation of increased beta phase in PP after e-beam radiation could be as follows. Irradiation causes chain scission in (i) lamellar regions or (ii) in amorphous interlamellar regions. Radiation occurred in several steps, 30 kGy each step. In each irradiation step, the material receives energy, temperature increases and the movement of the molecules is easier. Shorter chains have higher mobility which enables them to rearrange into a more stable crystalline state. However, the space for crystal formation is greatly restricted. Therefore, the chains rearrange only into a less stable $\beta$-phase. Further change into a more stable $\alpha$-form during further irradiation is restricted owing to the limited space and limited movement in the original interlamellar region.

Data accessibility. Data can be accessed in Dryad: https://doi.org/10.5061/dryad.dv41ns1wx [51].
Authors' contributions. P.S., main author, was involved in experiment design, data evaluation and writing; K.T. in experiments, data evaluation and writing; K.S. in results explanation; D.S. in English and introduction; T.O. was involved in results explanation
Competing interests. We declare we have no competing interests
Funding. We received no funding for this study.

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
