## [Peer Review File · Royal Society Open Science]

Review History

RSOS-202250.R0 (Original submission)

Review form: Reviewer 1

Is the manuscript scientifically sound in its present form?

Yes

Are the interpretations and conclusions justified by the results?

Yes

Is the language acceptable?

No

Do you have any ethical concerns with this paper?

Yes

Have you any concerns about statistical analyses in this paper?

No

Recommendation?

Accept with minor revision (please list in comments)

Comments to the Author(s)

Some revisions need be made and the comments are stated as follows:

- 1) For a broad impact, the applications of polypropylene and high-density polyethylene should be introduced and the following papers should be cited: Efficient Solvent-Free Microwave Irradiation Synthesis of Highly Conductive Polypropylene Nanocomposites with Lowly Loaded Carbon Nanotubes, *ES Materials & Manufacturing*, 2020, 9, 21-33; <https://doi.org/10.1007/s42114-019-00080-0>; Synthetic of Polydivinylbenzene Block Hyperbranched Polyethylene Copolymers via Atom Transfer Radical Polymerization, *ES Materials & Manufacturing*, 2019, 4, 20-24; Polyethylene Glycol/Carbon Black Shape-Stable Phase Change Composites for Peak Load Regulating of Electric Power System and Corresponding Thermal Energy Storage, *Engineered Science*, 2020, 9, 24-35; <https://doi.org/10.1007/s42114-019-00116-5>; <https://doi.org/10.1007/s42114-020-00181-1>
- 2) The language needs be improved together with the writing. So many paragraphs need be combined together to form a paragraph.
- 3) The importance of the Crystallization should be introduced and the following examples need be introduced and the references to be cited: Effect of PLA Crystallization on the Thermal Conductivity and Breakdown Strength of PLA/BN Composites, *ES Materials & Manufacturing*, 2019, 3, 66-72
- 4) For the “using the Scherrer equation”, the following reference needs be cited: Effect of Phosphine Gas Conditions on Structural, Optical and Electrical Properties of Nc-Si:H Films Deposited by Cat-CVD Method, *ES Materials & Manufacturing*, 2020, 10, 52-59

Review form: Reviewer 2

Is the manuscript scientifically sound in its present form?

Yes

Are the interpretations and conclusions justified by the results?

Yes

Is the language acceptable?

No

Do you have any ethical concerns with this paper?

No

Have you any concerns about statistical analyses in this paper?

No

Recommendation?

Major revision is needed (please make suggestions in comments)

Comments to the Author(s)

In this work, the author mainly investigated the crystallizing behavior of polypropylene (PP) and high-density polyethylene (HDPE) before and after electron beam irradiation. Totally, this work is comprehensive and interesting. I would like to support the publication of this manuscript in Royal Society Open Science. However, the following concerns need to be addressed:

- 1) It is inadvisable to use measuring methods as Keywords, such as DSC, optical microscopy, XRD.

- 2) In Fig. 1b, the authors proved that the polyethylene molecular chains have been cross-linked after irradiation and the corresponding resistance to creep was gradually improved with an increasing irradiation dose. However, in DMA measurement (Fig. 3), storage modulus of the HDPE samples was almost the same before and after irradiation. This is unreasonable, since the elastic properties of molecular chains would be enhanced after the crosslinking and the corresponding storage modulus would also increase. Please confirm this test result and give a reasonable explanation.
- 3) In Fig. 9, the crystallization rate of pure PP and the sample irradiated by 30kGy was faster than that of the samples irradiated by 60, 90 and 120kGy. The author attributed this to "The memory of the original chain positions remains in the material and it makes the nucleation step easier" and "The chain scission or branching influenced the ability of chains to fold to a lamellar structure". Please cite the related literatures.
- 4) Page 10, "The crystallization was analyzed initially by DSC...Then the crystallization kinetics can be expressed as $\tau^{-1/2}$ " should be moved to the Experimental part.
- 5) The authors should deeply explain the reasons why the e-beam radiation can promote the formation of the beta β phase of polypropylene in detail.
- 6). Figs. 4(a) and (b), the two FT-IR spectra cannot be well distinguished. Please do not overlap them and just put them in a certain distance.
- 7) The conclusions part needs amendment. The author only showed the corresponding results of each measurement; the reviewer thinks it is necessary to add the explanation why does electron beam irradiation have different effects on PE and PP, since this is the focus for the manuscript.
- 8) The English language needs an obvious improvement.

Decision letter (RSOS-202250.R0)

Dear Professor Svoboda:

Title: Study of crystallization behavior of electron beam irradiated polypropylene and high-density polyethylene

Manuscript ID: RSOS-202250

The editor assigned to your manuscript has now received comments from reviewers. We would like you to revise your paper in accordance with the referee and Subject Editor suggestions which can be found below (not including confidential reports to the Editor). Please note this decision does not guarantee eventual acceptance.

Please submit your revised paper before 12-Feb-2021. Please note that the revision deadline will expire at 00.00am on this date. If we do not hear from you within this time then it will be assumed that the paper has been withdrawn. In exceptional circumstances, extensions may be possible if agreed with the Editorial Office in advance. We do not allow multiple rounds of revision so we urge you to make every effort to fully address all of the comments at this stage. If

deemed necessary by the Editors, your manuscript will be sent back to one or more of the original reviewers for assessment. If the original reviewers are not available we may invite new reviewers.

On behalf of the Subject Editor Professor Anthony Stace and the Associate Editor Professor Chaohua Cui.

RSC Associate Editor:
Comments to the Author:
(There are no comments.)

RSC Subject Editor:
Comments to the Author:
(There are no comments.)

Reviewers' Comments to Author:
Reviewer: 1

Comments to the Author(s)

Some revisions need be made and the comments are stated as follows:

- 1) For a broad impact, the applications of polypropylene and high-density polyethylene should be introduced and the following papers should be cited: Efficient Solvent-Free Microwave Irradiation Synthesis of Highly Conductive Polypropylene Nanocomposites with Lowly Loaded Carbon Nanotubes, *ES Materials & Manufacturing*, 2020, 9, 21-33;

<https://doi.org/10.1007/s42114-019-00080-0>; Synthetic of Polydivinylbenzene Block Hyperbranched Polyethylene Copolymers via Atom Transfer Radical Polymerization, *ES Materials & Manufacturing*, 2019, 4, 20-24; Polyethylene Glycol/Carbon Black Shape-Stable Phase Change Composites for Peak Load Regulating of Electric Power System and Corresponding Thermal Energy Storage, *Engineered Science*, 2020, 9, 24-35; <https://doi.org/10.1007/s42114-019-00116-5>; <https://doi.org/10.1007/s42114-020-00181-1>

2) The language needs be improved together with the writing. So many paragraphs need be combined together to form a paragraph.

3) The importance of the Crystallization should be introduced and the following examples need be introduced and the references to be cited: Effect of PLA Crystallization on the Thermal Conductivity and Breakdown Strength of PLA/BN Composites, *ES Materials & Manufacturing*, 2019, 3, 66-72

4) For the “using the Scherrer equation”, the following reference needs be cited: Effect of Phosphine Gas Conditions on Structural, Optical and Electrical Properties of Nc-Si:H Films Deposited by Cat-CVD Method, *ES Materials & Manufacturing*, 2020, 10, 52-59

Reviewer: 2

Comments to the Author(s)

In this work, the author mainly investigated the crystallizing behavior of polypropylene (PP) and high-density polyethylene (HDPE) before and after electron beam irradiation. Totally, this work is comprehensive and interesting. I would like to support the publication of this manuscript in Royal Society Open Science. However, the following concerns need to be addressed:

1) It is unadvisable to use measuring methods as Keywords, such as DSC, optical microscopy, XRD.

2) In Fig. 1b, the authors proved that the polyethylene molecular chains have been cross-linked after irradiation and the corresponding resistance to creep was gradually improved with an increasing irradiation dose. However, in DMA measurement (Fig. 3), storage modulus of the HDPE samples was almost the same before and after irradiation. This is unreasonable, since the elastic properties of molecular chains would be enhanced after the crosslinking and the corresponding storage modulus would also increase. Please confirm this test result and give a reasonable explanation.

3) In Fig. 9, the crystallization rate of pure PP and the sample irradiated by 30kGy was faster than that of the samples irradiated by 60, 90 and 120kGy. The author attributed this to “The memory of the original chain positions remains in the material and it makes the nucleation step easier” and “The chain scission or branching influenced the ability of chains to fold to a lamellar structure”. Please cite the related literatures.

4) Page 10, “The crystallization was analyzed initially by DSC... Then the crystallization kinetics can be expressed as $\tau^{-1/2}$ ” should be moved to the Experimental part.

5) The authors should deeply explain the reasons why the e-beam radiation can promote the formation of the beta β phase of polypropylene in detail.

6). Figs. 4(a) and (b), the two FT-IR spectra cannot be well distinguished. Please do not overlap them and just put them in a certain distance.

7) The conclusions part needs amendment. The author only showed the corresponding results of each measurement; the reviewer thinks it is necessary to add the explanation why does electron beam irradiation have different effects on PE and PP, since this is the focus for the manuscript.

8) The English language needs an obvious improvement.

Author's Response to Decision Letter for (RSOS-202250.R0)

See Appendix A.

RSOS-202250.R1 (Revision)

Review form: Reviewer 1

Is the manuscript scientifically sound in its present form?

Yes

Are the interpretations and conclusions justified by the results?

Yes

Is the language acceptable?

Yes

Do you have any ethical concerns with this paper?

No

Have you any concerns about statistical analyses in this paper?

No

Recommendation?

Accept as is

Comments to the Author(s)

It can be accepted for publishing.

Review form: Reviewer 2

Is the manuscript scientifically sound in its present form?

Yes

Are the interpretations and conclusions justified by the results?

Yes

Is the language acceptable?

Yes

Do you have any ethical concerns with this paper?

No

Have you any concerns about statistical analyses in this paper?

No

Recommendation?

Accept as is

Comments to the Author(s)

The revisions are basically OK.

Decision letter (RSOS-202250.R1)

Dear Professor Svoboda:

Title: Study of crystallization behavior of electron beam irradiated polypropylene and high-density polyethylene
Manuscript ID: RSOS-202250.R1

It is a pleasure to accept your manuscript in its current form for publication in Royal Society Open Science. The chemistry content of Royal Society Open Science is published in collaboration with the Royal Society of Chemistry.

On behalf of the Subject Editor Professor Anthony Stace and the Associate Editor Professor Chaohua Cui.

RSC Associate Editor:
Comments to the Author:
(There are no comments.)

RSC Subject Editor:
Comments to the Author:
(There are no comments.)

Reviewer(s)' Comments to Author:
Reviewer: 1

Comments to the Author(s)
It can be accepted for publishing.

Reviewer: 2

Comments to the Author(s)
The revisions are basically OK.

Appendix A

Dear Reviewer # 1:

Thank you very much for your valuable comments.

Reviewer: 1

Some revisions need be made and the comments are stated as follows:

1) For a broad impact, the applications of polypropylene and high-density polyethylene should be introduced and the following papers should be cited:

Efficient Solvent-Free Microwave Irradiation Synthesis of Highly Conductive Polypropylene Nanocomposites with Lowly Loaded Carbon Nanotubes, ES Materials & Manufacturing, 2020, 9, 21-33; <https://doi.org/10.1007/s42114-019-00080-0>;

Synthetic of Polydivinylbenzene Block Hyperbranched Polyethylene Copolymers via Atom Transfer Radical Polymerization, ES Materials & Manufacturing, 2019, 4, 20-24;

Polyethylene Glycol/Carbon Black Shape-Stable Phase Change Composites for Peak Load Regulating of Electric Power System and Corresponding Thermal Energy Storage, Engineered Science, 2020, 9, 24-35; <https://doi.org/10.1007/s42114-019-00116-5>;

<https://doi.org/10.1007/s42114-020-00181-1>

We have inserted these three papers into the References numbers 1, 2 and 3.

2) The language needs be improved together with the writing. So many paragraphs need be combined together to form a paragraph.

For the language improvement we have used paid service of a native speaker from UK, Mr. David Catto, who lives in Luhacovice, Czech Republic.

Many paragraphs were combined together to form larger paragraphs.

3) The importance of the Crystallization should be introduced and the following examples need be introduced and the references to be cited:

Effect of PLA Crystallization on the Thermal Conductivity and Breakdown Strength of PLA/BN Composites, ES Materials & Manufacturing, 2019, 3, 66-72

The paper is now in References, number 8.

4) For the “using the Scherrer equation”, the following reference needs be cited:

Effect of Phosphine Gas Conditions on Structural, Optical and Electrical Properties of Nc-Si:H Films Deposited by Cat-CVD Method, ES Materials & Manufacturing, 2020, 10, 52-59

This paper was inserted into References, number 46.

Dear Reviewer #2:

Thank you very much for your time and valuable comments to improve the article.

Reviewer: 2

In this work, the author mainly investigated the crystallizing behavior of polypropylene (PP) and high-density polyethylene (HDPE) before and after electron beam irradiation. Totally, this work is comprehensive and interesting. I would like to support the publication of this manuscript in Royal Society Open Science. However, the following concerns need to be addressed:

1) It is unadvisable to use measuring methods as Keywords, such as DSC, optical microscopy, XRD.

We have changed the keywords per your recommendation.

Keywords: polymer; irradiation; crystallization; spherulite; beta-phase

2) In Fig. 1b, the authors proved that the polyethylene molecular chains have been cross-linked after irradiation and the corresponding resistance to creep was gradually improved with an increasing irradiation dose. However, in DMA measurement (Fig. 3), storage modulus of the HDPE samples was almost the same before and after irradiation. This is unreasonable, since the elastic properties of molecular chains would be enhanced after the crosslinking and the corresponding storage modulus would also increase. Please confirm this test result and give a reasonable explanation.

We have looked at the raw data and confirmed the validity. There is only a small increase in storage modulus in temperature range 60-120°C (see newly added Fig. 3b) and small decrease in tan delta (see newly added Fig. 3c). The crosslinking was done to a very small extent (compared to sulfur crosslinking in rubber industry). Therefore, the change in storage modulus (up) and tan delta (down) is only very moderate. The creep test above the melting point (shown in Fig. 1b) is much more sensitive to such small level of crosslinking.

3) In Fig. 9, the crystallization rate of pure PP and the sample irradiated by 30kGy was faster than that of the samples irradiated by 60, 90 and 120kGy. The author attributed this to “The memory of the original chain positions remains in the material and it makes the nucleation step easier” and “The chain scission or branching influenced the ability of chains to fold to a lamellar structure”. Please cite the related literatures.

We have added 4 papers (numbers 35, 36, 37, 38 in the References) concerning memory effects and 5 papers showing the influence of molecular weight and branching on crystallization kinetics (numbers 21, 27, 29, 32, 33 in the References).

4) Page 10, “The crystallization was analyzed initially by DSC...Then the crystallization kinetics can be expressed as $\tau^{1/2-1}$ ” should be moved to the Experimental part.

The whole paragraph was moved to the Experimental part.

5) The authors should deeply explain the reasons why the e-beam radiation can promote the formation of the beta β phase of polypropylene in detail.

We have added text concerning this topic before conclusion citing 3 different authors (Nedkov et al. 2004, Manas et al. 2013, Bhateja et al. 1995) plus we have added a paragraph into Conclusion.

Explanation of increased beta phase in polypropylene after e-beam radiation could be this. Irradiation causes chain scission in (a) lamellar regions or (b) in amorphous interlamellar regions. Radiation occurred in several steps, 30 kGy each step. In each irradiation step the material receives energy, temperature increases and the movement of the molecules is easier. Shorter chains have higher mobility which enables them to rearrange into more stable crystalline state. However, the space for crystal formation is greatly restricted. Therefore, the chains rearrange only into less stable β -phase. Further change into more stable α -form during further irradiation is restricted due to the limited space and limited movement in original interlamellar region.

6). Figs. 4(a) and (b), the two FT-IR spectra cannot be well distinguished. Please do not overlap them and just put them in a certain distance.

We have changed Figs. 04a and 04b per your instruction. The inset in Fig. 04a shows the original data (not shifted).

7) The conclusions part needs amendment. The author only showed the corresponding results of each measurement; the reviewer thinks it is necessary to add the explanation why does electron beam irradiation have different effects on PE and PP, since this is the focus for the manuscript.

The conclusion contains now the following amendment:

Our results of crystallization kinetics influenced by irradiation show a decrease in kinetics for both: PE and PP, even though the causes are completely different because PP (in contrast to PE) contains a tertiary carbon atom in the main chain. In the case of PE there is an increase in molecular weight, branching and cross-linking. In the case of PP we found a decrease in the following: molecular weight, bulk crystallization kinetics, spherulitic crystallization kinetics, and the number of spherulites. These results correspond well with other researchers [33] who demonstrated a “U” shape of $\tau_{0.01}$ which is inversely proportional to crystallization kinetics as a function of Mw. In lower molecular weight range the decrease in Mw

leads to higher $\tau_{0.01}$ (or slower crystallization). In higher molecular weight range the increase in Mw leads also so slower crystallization explained by reptation model [29].

8) The English language needs an obvious improvement.

For the language improvement we have used paid service of a native speaker from UK, Mr. David Catto, who lives in Luhacovice, Czech Republic.